# Three-dimensional observations of the electric field distribution of variable frequency microwaves, and scaling-up organic syntheses

Satoshi Horikoshi [1✉], Haruka Mura[1] & Nick Serpone[2]

Variable Frequency Microwave (VFM) radiation provides a solution to the inhomogeneity of the electric field in the cavity, which has long led to a decline in the reliability of microwave chemical data and its industrial utilization. Herein, we report in-situ three-dimensional experimental measurements of the electric field's uniform distribution of VFMs within a multimode cavity under high power conditions, and their subsequent comparison to Fixed Frequency Microwaves (FFM) that could only be assessed earlier through theoretical analysis. We also examine the consequences of changes in VFM irradiation conditions and elucidate the threshold at which VFM irradiation might prove beneficial in syntheses. With an ultimate focus on the use of VFM microwave radiation toward industrial applications, we carried out an effective synthesis of 4-methylbyphenyl (4-MBP) in the presence of palladium (the catalyst) supported on activated carbon particulates (Pd/AC), and revisited two principal objectives: (a) the effective suppression of discharge phenomena (formation of hot spots), and (b) synthesis scale-up using a 5-fold increase in sample quantity and a 7.5-fold larger reactor size (diameter) than otherwise used in earlier studies.

[1] Department of Materials and Life Sciences, Faculty of Science and Technology, Sophia University, Chiyodaku, Tokyo, Japan. [2] PhotoGreen Laboratory, Dipartimento di Chimica, Università di Pavia, Pavia, Italy. ✉email: horikosi@sophia.ac.jp

Two papers published in 1989 proposed an innovative methodological concept in carrying out organic syntheses[1,2], which can best be described from the old adage *learn from the old, but know the new* that involved the use of a domestic microwave oven as the heat source to carry out chemical syntheses. Considerable interest followed soon thereafter from the Chemistry Community in the use of microwave energy as an alternative heating method because of some advantages this original technology based on microwaves might offer over conventional heating methods. Currently, the use of microwaves as a heating source to conduct chemical reactions is no longer uncommon. Indeed, subsequent to the use of domestic microwave ovens in the early years of research into microwave chemistry, microwave equipment with user interfaces has become available that can control automatically both the temperature and the microwaves with regard to frequency range and output power. Users with scarce knowledge of the physical manifestations of electromagnetic waves and custom-made microwave equipment can easily perform chemical syntheses now with modern microwave equipment in a manner that parallels the use of a domestic microwave oven.

The use of microwaves in the chemical industry is expected to increase; however, this will require continued fundamental research into this otherwise promising microwave technology so as to achieve process growth owing to rapid, volumetric, and selective heating by microwaves as opposed to conventional heating. These unique abilities of microwave heating have been summarized in several articles, and related reactions have been designated as Green Chemistry processes[3–5]. In this regard, different types of reactions have been summarized in a technical book co-edited by de la Hoz and Loupy[6]. The possibility of scale-up of some reactions has also been examined and scaled-up reactor designs have also been reported[7–10]. The growth of examples of organic reactions in microwave chemistry has been nearly exponential, although some published studies have been less than accurate. For example, Kappe[11] pointed out that the effect of accelerating reactions in microwave chemistry has been rather problematic with regard to temperature measurements within the reaction solution. For instance, applying the known Arrhenius equation for determining chemical reaction rates, an erroneously measured temperature can lead to significant errors in reaction rates. When electromagnetic waves from microwave radiation are used as a heat source in performing chemical reactions, the inhomogeneity in the temperature distribution cannot be removed by simple stirring[12]. To obtain temperature homogeneity necessitates uniformity of the electric field (*E*-field) distribution during microwave heating, which will require some adjustments in various parameters such as (i) microwave penetration depth in samples, (ii) designs of microwave cavities, (iii) types of reactors and (iv) stirring of the reaction solutions or otherwise the microwave mode. Making these adjustments in microwave chemistry, however, requires considerable experience, not least of which is the inhomogeneous distribution of the *E*-field intensity that can also lead to additional problems.

Reactions carried out in the presence of solid catalysts can induce selective heating of the catalyst's surface by the microwaves and thus bring the reaction field to a higher temperature relative to the bulk solution temperature[13]. This advantage has been explored in various fields[14–17]. For instance, in the seminal review by Priecel and Lopez-Sanchez[18], the authors appraised the application of microwave reactors in the catalytic transformation of biomass and biomass-derived molecules with a particular emphasis on heterogeneous catalysis. In addition, the interaction between solid catalysts and microwaves has also been investigated for their size effects[19]. As well, the induced selective heating of the catalyst in microwave chemistry has afforded knowledge of

chemical reaction kinetics that cannot otherwise be achieved with existing heat sources. Evidence for microwave selective heating was reported by Ano and coworkers[20] for the Pt catalyst surface using in situ extended X-ray absorption fine structure (EXAFS). These same authors also suggested that the reducing power of the Pt catalyst surface is enhanced by microwaves as demonstrated by operando X-ray absorption near edge structure (XANES) spectroscopy. No doubt, industry will pay more attention to the use of these technologies.

The above notwithstanding, however, chemical processes carried out in the presence of solid catalysts display problems caused by the inhomogeneity of the microwaves' *E*-field. Performing reactions under such conditions can lead to the generation of hot spots (microwave-generated non-equilibrium temperature microscopic spots) between solid catalyst particulates because of the concentration of the microwaves' *E*-field within the microwave reactor[21,22]. Such generation occurs when a substrate or solvent (*e.g.*, a non-polar substance) with low dielectric loss is irradiated with microwaves in a dispersed catalyst or catalyst support (*e.g.*, activated carbon) that absorbs a significant quantity of microwave energy. The existence of hot spots has been demonstrated by means of a high-speed camera on a Pd/AC catalyst surface during a Suzuki-Miyaura cross-coupling reaction taking place in toluene solvent[23]. As a result, the Pd catalyst nanoparticles dispersed on the surface of AC agglomerates caused the catalytic activity to be reduced considerably. Consequently, hot spots need to be suppressed, which can be achieved by homogenizing the microwaves' electric field throughout the reaction space.

The variable frequency microwave (VFM) technology was developed in 1991 at the Oak Ridge National Laboratories[24,25], a technology known for its theoretically uniform microwave *E*-field distribution for irradiation purposes. Recent relevant research[26] has shown that the discharge generated on a solid catalyst within the reactor located in a single-mode cavity can be suppressed using microwaves whose frequency is continuously swept at high speed (variable frequency microwaves: VFM; range, 5.85–6.65 GHz). In this regard, an actual synthesis of 4-methylbiphenyl (4-MBP) carried out in the presence of a Pd/AC catalyst as well as in the dehydrogenation of methylcyclohexane using a Pt/AC catalyst led to 6–10 times greater yields of 4-MBP, and more than 3 times higher dehydrogenation than those under fixed frequency microwave (FFM) conditions. This was attributed to the suppression of microwave standing waves resulting from improved uniformity of the *E*-field. Other reports have shown improvements of the *E*-field uniformity under VFM irradiation in the vulcanization of rubber[27], in adhesives[28], and in nanoparticle printing[29], thus forewarning the effectiveness of VFM irradiation in chemical reactions. However, although the distribution of the *E*-field has been established theoretically through simulations, an experimental distribution of the *E*-field intensity inside a multimode cavity under VFM irradiation has, to our knowledge, not been investigated. For that reason, we found it necessary to ascertain whether the VFM irradiation technology can suppress uneven heating and formation of hot spots caused by the non-uniformity of the *E*-field, which have hitherto been serious issues in microwave heating.

Accordingly, the objective of the present study was to examine experimentally the effectiveness of VFM irradiation by in-situ three-dimensional observation of the *E*-field within a multimode cavity that earlier could only be accessed through theoretical simulations. This was achieved by (i) probing the *E*-field distributions of the VFM and FFM microwaves in the multimode cavity first, followed by (ii) correlating the frequency conditions and *E*-field distribution upon VFM irradiation, and finally by (iii) exploring the characteristics of temperature changes of liquid and

solid/liquid samples under VFM irradiation. These features were then systematized toward (iv) carrying out the synthesis of 4-methylbiphenyl (4-MBP) *via* the Suzuki-Miyaura cross-coupling reaction in the presence of a solid Pd catalyst supported on activated carbon (AC) particulates in multimode cavities under scaled-up conditions.

## Results and discussion
**Difference(s) between VFM and FFM methods**. The distribution of the *E*-field intensity with respect to the horizontal plane inside the cavity was investigated using VFM radiation with a microwave power output of 18 Watts, and with the frequency being swept continuously from 5.85 to 6.65 GHz in 0.1 s intervals. Comparison of the characteristics of the *E*-field intensity of the VFM radiation vis-à-vis the features of the FFM radiation was achieved using the same apparatus also with 18-Watt microwaves. Nonetheless, prior to this, a preliminary experiment was conducted to evaluate the performance of microwaves generated from the VariWave apparatus (Lambda Technologies Inc.). The *E*-field intensity was observed at 5.85–6.65 GHz under VFM conditions with the frequency variation from 5.85 to 6.65 GHz being within ± 0.02 GHz. Interestingly, no frequency variation was experienced under FFM microwave conditions at 5.85 GHz and 6.65 GHz. The *E*-field intensity was measured at positions of 1 cm, 7 cm, and 14 cm above the bottom of the cavity and measured every 2 cm from the center up to 10 cm toward the $E_1$, $E_2$, and $E_3$ directions that covered an area of $20 \times 20$ cm$^2$ as illustrated in Fig. 1a.

Figure 1b, c display the intensity of the *E*-field with respect to the plane at each measurement point. The average value of the *E*-field intensity (kV m$^{-1}$) of all measurements was calculated as noted earlier; the difference between the estimated average value of the *E*-field intensity from Eq. 2 (see Experimental section) and the actual measured value at each measurement point was plotted along the Y axis of the graphics; a value of 100% on the Y axis represents the average value—in other words, the actual values in the plots were normalized to the estimated average values. The X and Z axes define the bottom surface of the cavity, with the center representing the intersection of the X-axis and the Z-axis. A comparison of the *E*-field intensity under VFM irradiation and FFM irradiation on a horizontal plane at a height of 14 cm from the cavity bottom shows that under VFM (Fig. 1b–i) the *E*-field intensity is uniformly distributed. In contrast, the *E*-field intensity measured at the same position under FFM (5.85 GHz) irradiation deviated greatly from the average value (see Fig. 1c–i), as indicated by the standard deviation of the *E*-field intensity for each measurement (Fig. 1d). To systematize the results, the variation of the *E*-field intensity distribution based on a representative value of the standard deviation is shown (Fig. 1d), in which the variation in total values of the population was calculated using STDEV.P on MS-Excel. Results revealed that the standard deviation under VFM irradiation was 0.19, while under FFM irradiation it was 0.32. This suggests that the uniformity of the *E*-field was improved by 41% when using VFM microwaves.

A similar measurement of the *E*-field intensity at a position 7 cm from the bottom of the cavity under VFM (Fig. 1b-ii) versus FFM irradiation (Fig. 1c-ii) also revealed a non-insignificant contrast as ascertained by the standard deviations: 0.15 for VFM and 0.28 for FFM radiations (Fig. 1d). Once again, there was a 46% improvement of the *E*-field uniformity under VFM conditions.

Figure 1b-iii illustrates the *E*-field intensity under VFM irradiation on the horizontal plane at a height of 1 cm above the cavity bottom. The *E*-field intensity distribution under VFM at 1 cm is less uniform than that at 7 cm (Fig. 1b-ii). In comparison,

the uniformity distribution of the *E*-field intensity under FFM irradiation (Fig. 1c-iii) tended to decrease relative to those at 14 cm and 7 cm. The standard deviation of the *E*-field intensity was 0.23 for VFM irradiation (Fig. 1d), and 0.40 for FFM irradiation. Here also, VFM irradiation led to a significant 43% improvement of the uniformity of the *E*-field *via-à-vis* FFM irradiation.

In order to ascertain and elucidate the above experimental results from the heating experiments, we proceeded to visualize the changes in heating using Thermopaint (Narika Co., Japan; 0.15 g), which changes color from blue to pink at 40 °C. To do so, 7 mL of the Thermopaint was mixed with agar dissolved in 300 mL of ion-exchanged water that was then poured into a 21.0 × 29.0 cm$^2$ (4.0 cm high) white plastic vat to solidify the agar. The vat was subsequently placed horizontally at a position 7 cm from the bottom of the cavity (see Fig. 2a). In this experiment, as the water was detained by the agar, heat conduction by convection was avoided. Explicitly, only the positions heated by the microwaves turned pink, so that this color change from blue to pink equates with a high *E*-field intensity. We also observed that after microwave irradiation (Fig. 2c), the blue agar (Fig. 2b) changed to an overall light pink color after 70 s under 180-Watt VFM microwave irradiation (Fig. 2d), thereby demonstrating uniform heating of the agar under VFM conditions. Nonetheless, even with VFM irradiation, there are darker and lighter areas in the pink color change. In contrast, only a minor fraction of the agar/Thermopaint turned dark pink under FFM irradiation with a significantly larger fraction remaining blue. Clearly then, irradiation with VFM microwaves led to uniform heating over a wide area, from which we infer that the *E*-field intensity was considerably uniform.

**VFM conditions and E-field distribution**. Next, we examined the characteristics of the uniformity of the *E*-field intensity distribution for each condition of VFM irradiation. To do so, we used standard VFM conditions (same conditions as in Fig. 1b-ii: cavity height, 7 cm; sweep speed, 0.1 s; frequency sweep width, 5.85 to 6.65 GHz; microwave output power, 18 Watts), and proceeded to investigate the *E*-field intensity uniformity by changing (a) the frequency sweep speed, (b) the sweep width, and (c) the microwave power. As well, we measured the *E*-field intensity using (d) different microwave chemical equipment in order to compare a magnetron microwave generator with a semiconductor microwave generator.

*Frequency sweep speed*. The repetition time of the frequency sweep (sweep rate) in VFM irradiation was changed from 0.1 s to 0.7 s, 1.5 s, and 3.0 s in order to examine the uniformity of the *E*-field intensity distribution in the multimode cavity. Other conditions were consistent with those in Fig. 1b-ii. When the sweep speed was extended to 0.7 s (Fig. 3a-i), the inhomogeneity of the *E*-field intensity became larger. Extending the sweep speed to 1.5 s (Figs. 3a-ii) and 3.0 s (Fig. 3a-iii) caused the inhomogeneity of the *E*-field intensity to be prolonged in proportion to the sweep speed. The standard deviation of each measured value of the *E*-field intensity was ca. 0.15 at 0.1 s, changing to 0.22 at 0.7 s, 0.23 at 1.5 s, and 0.36 at 3.0 s (Fig. 3a-iv). In comparison, the standard deviation was 0.28 under FFM irradiation (5.85 GHz; frequency sweep speed = 0; Fig. 3a-iv), which indicates that the *E*-field inhomogeneity was largest under the 3.0-s irradiation period. Regardless, why did the deviation from the average value of the *E*-field become larger when sweeping for 3.0 s rather than when frequency sweeping for 0 s? Perhaps, shorter wavelength microwaves other than those at 5.85 GHz were generated up to 6.65 GHz under these conditions, which may have resulted in the generation of *E*-field variation within a smaller region.

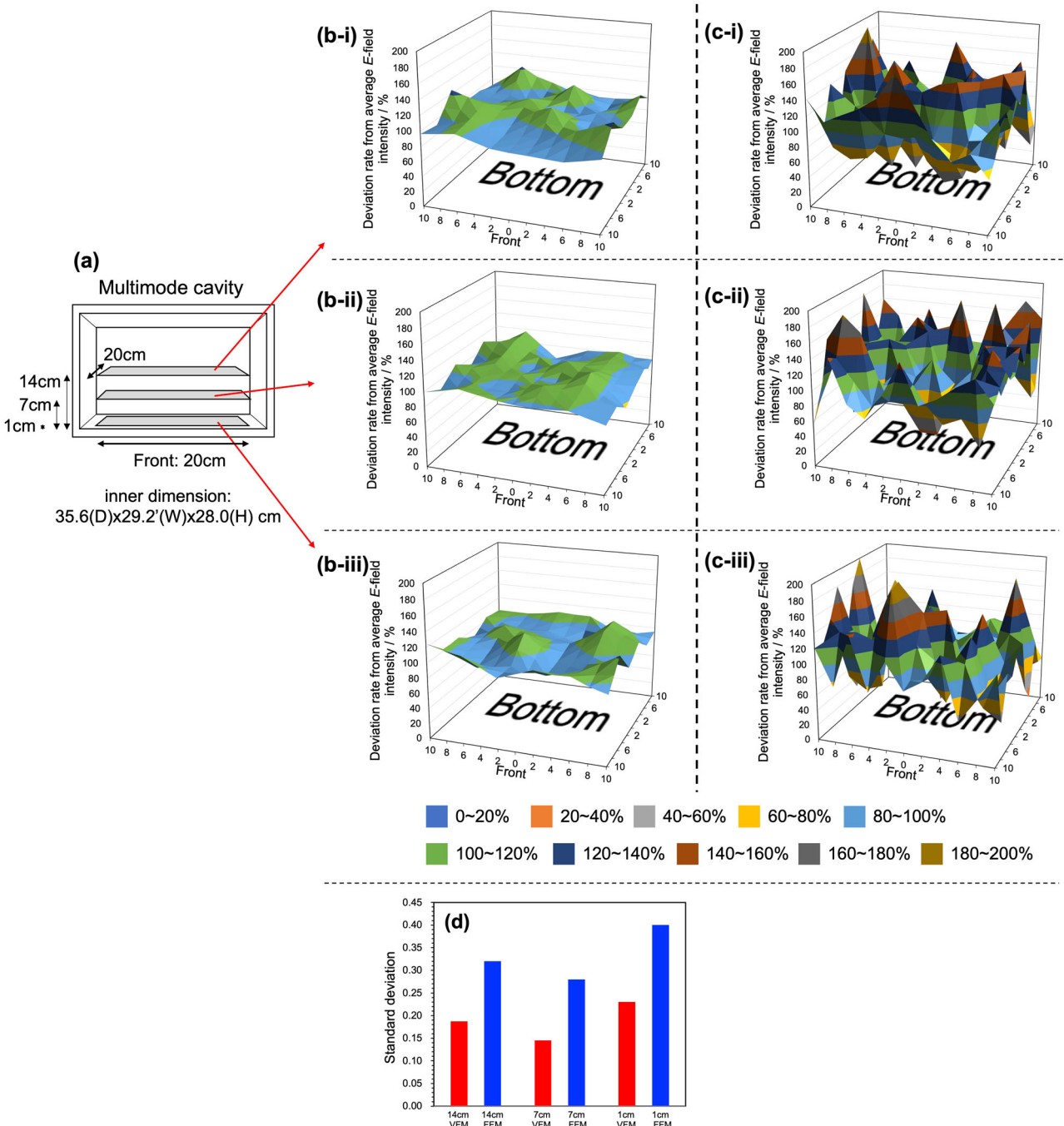

**Fig. 1 Comparison of contour plots and standard deviations of the distribution of the *E*-field intensity with changes under variable frequency microwave (VFM) irradiation and fixed frequency microwave (FFM) irradiation on the horizontal plane in the microwave multimode cavity. a** Image of the measurement position in the multimode cavity, **b** VFM irradiation, **c** FFM irradiation, (-i) position 14 cm from the bottom, (-ii) position 7 cm from the bottom, (-iii) position 1 cm from the bottom, **d** summary of deviation from the average value of the *E*-field intensity by the standard deviations.

*Frequency sweep width.* The frequency sweep width was varied from 0.8 GHz to 0.7 GHz (range 5.85–6.55 GHz; Fig. 3b-i), to 0.6 GHz (5.85–6.45 GHz; Fig. 3b-ii), to 0.4 GHz (5.85–6.25 GHz; Fig. 3b-iii) and to 0.2 GHz (5.85–6.05 GHz; Fig. 3b-iv) to investigate the distribution of the *E*-field intensity in the cavity. The deviation from the average value of the *E*-field intensity in the plane increased as the sweep width of the frequency was narrowed. This is because the narrower the sweep width of the swept frequency is, the narrower is the wavelength and the closer is the sweep to FFM irradiation. The standard deviation of each measurement with respect to the *E*-field intensity was then calculated

(Fig. 3b-v): the standard deviation was 0.15 at a sweep width of 0.8 GHz, which changed to 0.20 at 0.7 GHz, to 0.24 at 0.6 GHz, to 0.27 at 0.4 GHz and to 0.28 at 0.2 GHz. The standard deviation under FFM irradiation (5.85 GHz) was 0.28, indicating that the *E*-field misalignment rate was not different from that of FFM at a sweep width of about 0.2 GHz.

*Microwave power.* The *E*-field intensity with respect to the plane was also examined when the microwave power was doubled under the same conditions as those in Fig. 1b-ii. We observed, however, that the rate of displacement of the *E*-field intensity did

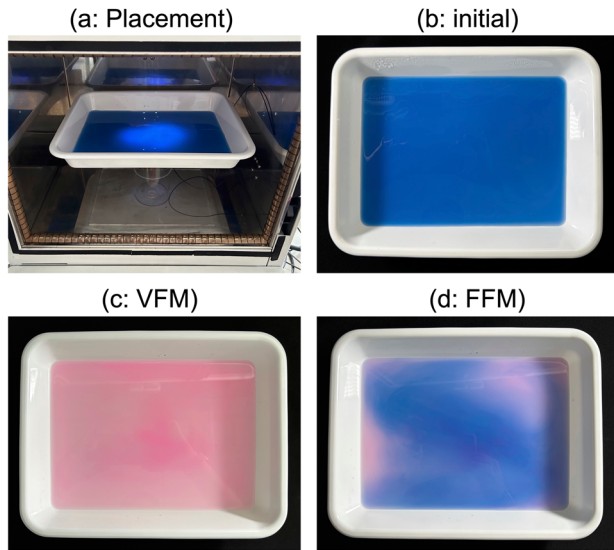

**Fig. 2 Photograph of the heating distribution in agar containing Thermopaint under VFM and FFM irradiation. a** photograph of the sample placed 7 cm from the bottom of the cavity, **b** photograph of a container with agar containing Thermopaint before microwave irradiation, **c** photograph of agar containing Thermopaint after VFM irradiation for 70 s and **d** photograph of agar containing Thermopaint after 70 s of FFM irradiation.

not double on increasing the microwave power from 18 Watts to 36 Watts (Fig. 3c-i); the calculated deviation from the average value was 0.15 (18 Watts) versus 0.22 (36 Watts; Fig. 3c-iii).

*Use of different microwave apparatus.* As the final task, we subsequently compared a semiconductor generator with VFM irradiation versus a commercially available magnetron generator with FFM microwave irradiation (FlexiWAVE; 2.45 GHz, Milestone General Co.)—see Fig. 3c-ii versus Fig. 3c-iii. The E-field was measured at a height of 7 cm from the bottom of the microwave chemical reactor cavity; moreover, the E-field intensity was also measured at 10 cm from the center of the microwave chamber in all three directions. Preliminary experiments showed that the minimal stable microwave oscillation output of the microwave chemical reaction (FFM) apparatus was 30 Watts; in our case we used a 36-Watt microwave power output. The standard deviation was 0.22 for VFM irradiation using a semiconductor generator with the same output power (36 Watts), and 1.96 under FFM (2.45 GHz) irradiation using the magnetron generator (Fig. 3c-iii); thus, VFM irradiation achieved a nearly 9-fold greater E-field uniformity. In other words, by changing from the mechanical mode stirrer to a frequency sweep (electrical mode stirrer), it is possible to greatly suppress disturbances in the E-field; note that this difference is not just a matter of electric field agitation. The reason for this lies in the quality of the microwaves generated by the magnetron generator. Figure 4 shows the spectrum of frequency *versus* E-field intensity in the multimode cavity obtained when the E-field intensity was measured by the optical electric field sensor in the FlexiWAVE apparatus. Clearly, the frequencies are dispersed from 2.45 GHz, which causes even greater fluctuations in the E-field intensity.

Using a microwave device with lower performance, namely a commercial domestic microwave oven (IRIS Ohyama Inc., Japan; IMB-T178-W; 2.45 GHz; maximal output 650 Watts), the calculated standard deviation of the measured E-field distribution was 4.2, nearly 23 times greater than the 0.18 for VFM irradiation using a semiconductor generator, thus indicating a non-uniform E-field distribution when using the domestic microwave source.

In comparison with a domestic microwave oven, the flexiWAVE apparatus can provide a sufficiently uniform E-field intensity distribution, which furthers our understanding on how the uniformity of the E-field intensity of VFM irradiation with the semiconductor-type transmitter and the sweeping of the microwave frequency is greater than that delivered by a domestic microwave oven.

*Summary of VFM irradiation characteristics.* (**I**) VFM irradiation improved the uniformity of the E-field intensity by 41% or more compared to FFM irradiation. Under VFM irradiation, the frequency interval of 0.8 GHz from 5.85 to 6.65 GHz was continuously swept in 4096 steps in 0.1 s intervals with one step being 0.195 MHz. For example, Fig. 5a shows the superposition of the sinusoidal waves oscillating from λ = 5.12 cm (5.85 GHz) to λ = 4.51 cm (6.65 GHz) in steps of 10 MHz used for VFM irradiation. As the microwaves advance, the wavelength shift increases, thereby filling the gap between the wavelengths. Note that the actual VFM irradiation synthesizes 50 times more microwave frequencies; in this way, the E-field of each microwave with a small wavelength shift compresses the space (see Fig. 5a). On the other hand, resonance is inhibited because there are no microwaves present at the same position, so that the microwaves in the cavity exist uniformly. As a matter of course, however, the sinusoidal wave of the 5.85 GHz frequency (Fig. 5b) has a place where a standing wave can be generated owing to the half wavelength at λ = 2.56 cm. In other words, dense and sparse portions of the E-field are generated.

(**II**) Even under VFM irradiation, it was not possible to provide a uniform E-field over the entire cavity. In fact, the uniformity of the E-field intensity decreased slightly depending on the position. In the vertical direction, the uniformity of the E-field decreased in the order: 7 cm > 14 cm > 1 cm. In particular, there was a tendency for the uniformity of the E-field intensity to decrease at locations close to the bottom surface and close to the wall surfaces inside the cavity cabinet. This is thought to be due to the reflected waves which are greatly affected by the proximity to the wall inside the cavity. As a consequence, even VFM irradiation is affected by the latter.

(**III**) With regard to the frequency sweep speed, the shorter the sweep repetition time is under VFM irradiation, the better it is. For instance, VFM irradiation became stronger than FFM irradiation above 3.0 s. As for the frequency sweep width, even around 0.4 GHz, there was almost no difference from FFM irradiation (5.85 GHz). In addition, a frequency sweep width of more than 0.7 GHz was necessary to reduce the standard deviation to less than 0.2. Insofar as microwave power is concerned, a change in power output when using VFM irradiation did not correlate with a change in the E-field intensity distribution. Importantly, compared to commercial microwave heating apparatus/equipment, VFM irradiation enhanced the leveling of the E-field intensity nearly 9-fold in the cavity. However, the semiconductor generator used in this research tends to be more expensive than the magnetron oscillator, which may in some cases limit its practical application. Nevertheless, it is generally known that the cost of semiconductor devices decreases exponentially depending on the number of products produced. This problem can be mitigated if the importance of the VFM method were recognized, and the number of users increased.

**Heating of pristine solvents and AC/solvent dispersions**. The characteristics of heating with VFM or FFM microwave radiation were probed using a polar solvent (1-hexanol), a non-polar solvent (toluene), and a microwave absorber (AC particles) in which heat generated by microwaves evolves. Before starting the

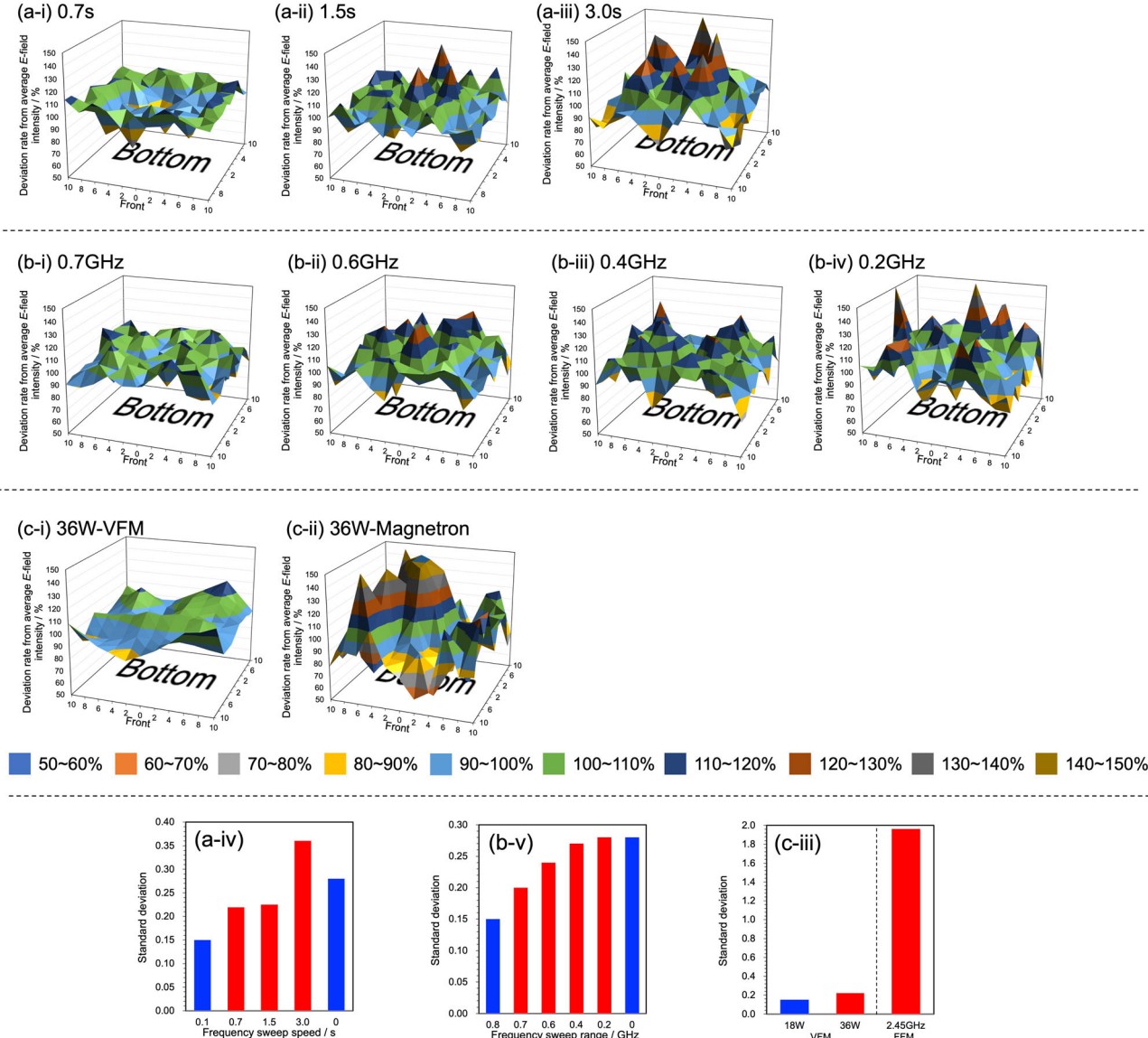

**Fig. 3 Comparison of contour plots of the distribution of the *E*-field intensity with changes in VFM irradiation conditions and the relevant standard deviations. a** comparison against frequency sweep speed (i: 0.7 s, ii: 1.5 s and iii: 3.0 s), **b** comparison against frequency sweep width (i: 0.7 GHz ii: 0.6 GHz, iii: 0.4 GHz, iv: 0.2 GHz), **c** comparison with (i) microwave power and (ii) magnetron generator. (**a**-iv, **b**-v and **c**-iii) summary of standard deviations from the average value of the *E*-field intensity.

experiment, however, it was vital to examine how the dielectric factors (*e.g.*, dielectric constant ε' (Fig. 6a); dielectric loss ε'' (Fig. 6b); and tan δ (Fig. 6c)) for 1-hexanol and toluene were affected by sweeping the microwave frequency from 5.85 GHz to 6.50 GHz; appropriate measurements were carried out using a network analyzer (Keysight Technologies E5071C with a Slim Form Probe Kit, Option 030). In the case of 1-hexanol, the values of dielectric loss and tan δ, which determine microwave heating, tended to decrease somewhat with increasing frequency.

The effect on the penetration depth was calculated from the results of the dielectric factors (see below and Fig. 6d). Note that the penetration depth (*Dp*) is the depth at which the microwaves pervade the material when the power flux falls to $1/e$ (= 36.8%) of its surface value; its magnitude was estimated from Eq. 1[30]. For example, the penetration depth of 5.85-GHz microwaves was about 5.4 cm for 1-hexanol and about 106 cm for toluene. Based on these results, a quartz test tube (height 15.0 cm) with an outer diameter of 2.7 cm and an inner diameter of 2.5 cm was used, a

size that allowed microwaves to irradiate adequately the center of the reactor.

$$Dp = \frac{\lambda}{4\pi} \left[ \frac{2}{\varepsilon' \left( \sqrt{1 + \left( \frac{\varepsilon''}{\varepsilon'} \right)^2} - 1 \right)} \right]^{\frac{1}{2}} \quad (1)$$

In view of the above requirements, 1-hexanol (7 mL) or toluene (7 mL) was placed in quartz test tubes and subsequently heated with VFM and FFM (5.85 GHz or 6.65 GHz) microwaves. The temperature profile of 1-hexanol (Fig. 7a) shows that when VFM (5.85–6.65 GHz) is used for 60 s, the temperature reaches about 102 °C, whereas under FFM irradiation at 5.85 GHz the temperature reaches 89 °C and at 6.65 GHz it is ca. 120 °C. Calculating the average value of the temperature change of these FFM irradiations, in comparison with the temperature change under VFM irradiation, yielded a theoretical average temperature of 105 °C after 60 s for FFM, similar to the temperature rise under VFM irradiation;

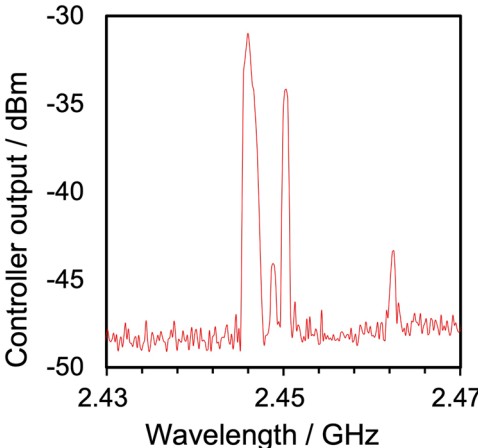

**Fig. 4 Spectrum of the *E*-field intensity *versus* frequency in the multimode cavity.** The microwave system used was Milestone General Co. flexiWAVE apparatus (2.45 GHz).

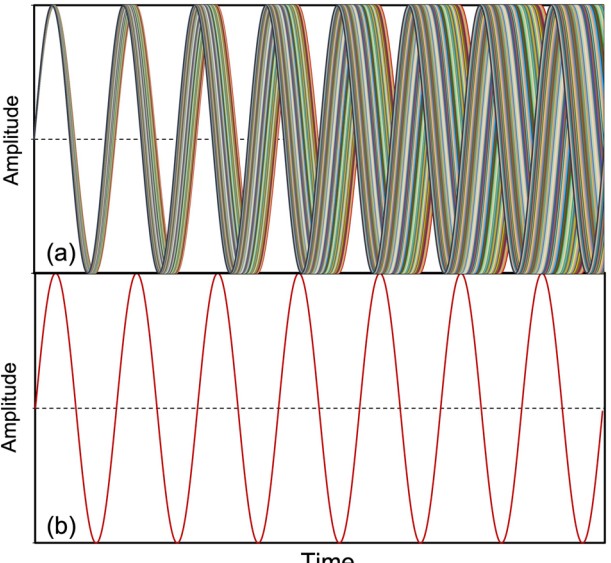

**Fig. 5 Superposition of the sinusoidal waves oscillating from 5.85 GHz to 6.65 GHz in steps of 10 MHz, and illustration of the sinusoidal wave at 5.85 GHz.** **a** Sinusoidal waves at 5.85 GHz to 6.65 GHz and **b** sinusoidal wave at 5.85 GHz.

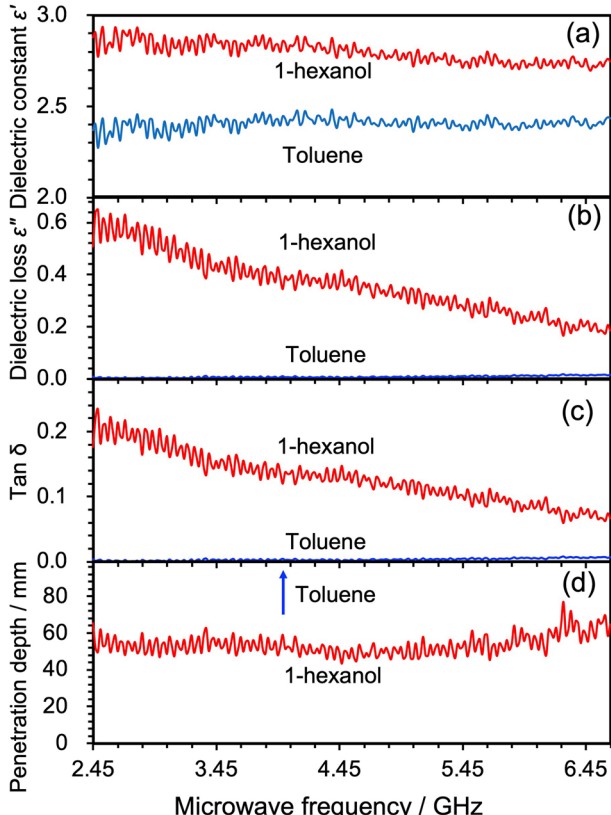

**Fig. 6 Effect of frequency sweeping on various dielectric factors.** **a** dielectric constant $\varepsilon'$, **b** dielectric loss $\varepsilon''$, **c** tan $\delta$, and **d** the penetration depth of 1-hexanol and toluene from 5.85 GHz to 6.50 GHz.

interestingly, the temperature change profile was also rather similar. On the other hand, the tendency of the dielectric loss (Fig. 6b) and tan $\delta$ (Fig. 6c) for 1-hexanol suggests that the heating efficiency was nearly independent of frequency. So perhaps the heating efficiency may depend on the *E*-field density. In this regard, the wavelength of 5.85-GHz microwaves is 5.12 cm, while the wavelength of 6.65-GHz microwaves is 4.51 cm, and the *E*-field density is about 12% higher. The inference on the dependence of the heating efficiency on the *E*-field density accords with an earlier report by Horikoshi and Serpone[31]. Next, we compared the temperature rise for toluene (Fig. 7b). Even with FFM and VFM, the temperature remained in the range 34–38 °C after heating for 60 s. Since the initial temperature of the toluene solvent was 28 °C, heating hardly progressed under any conditions, as also witnessed under VFM irradiation.

Next, we explored the effect of VFM and FFM irradiation on activated carbon powder (AC; particle size ca. 32 µm; 0.45 g) dispersed in either 1-hexanol (7 mL) or toluene (7 mL). The 1-

hexanol/AC dispersions were irradiated for 60 s with VFM, FFM (5.85 GHz), and FFM (6.65 GHz) microwaves. The temperature reached 90 °C for VFM, 71 °C for FFM (5.85 GHz), and 134 °C for FFM (6.65 GHz). The average FFM temperature was 103 °C after 60 s (Fig. 7c), more than 10 °C higher than under VFM irradiation. On the other hand, heating a toluene/AC dispersion under different microwave irradiation conditions, also after 60 s, reached a temperature of 42 °C under VFM, 37 °C under FFM (5.85 GHz), and 57 °C for FFM (6.65 GHz) (see Fig. 7d). The theoretical average temperature of the two FFMs was 47 °C after 60 s, a difference of more than 5 °C compared to VFM. Clearly, selective heating of the AC particulates led to an increase of the heating rate of toluene.

Although these results are not too far off those of the pure solvents, the inclusion of AC does shed some light on the difference between the theoretical mean temperature under FFM and the temperature under VFM irradiation. We expected the temperature change on the AC particles to be characteristically distinct for VFM irradiation and FFM irradiation. To ascertain the latter expectation, AC particles were spread onto a quartz petri dish (inner diameter: 4.0 cm; height: 1.7 cm), and then subjected to VFM and FFM heating, with the surface temperature of the AC particles measured no less than 3 times using a radiation thermometer; the temperature variation was < 2 °C. Under VFM irradiation, the temperature of pure AC particles reached 81 °C after 60 s (Fig. 7e), whereas the temperatures reached 50 °C and 134 °C, respectively, under FFM (5.85 GHz) and FFM (6.65 GHz) irradiation; the theoretical average temperature was 92 °C, some 6 °C higher than for VFM irradiation. According to Kim et al.[32], microwave heating of AC particulates is caused by Joule heating that results from the movement of electrons inside the AC in such a way that the heating efficiency increases with increase in the microwave frequency. However,

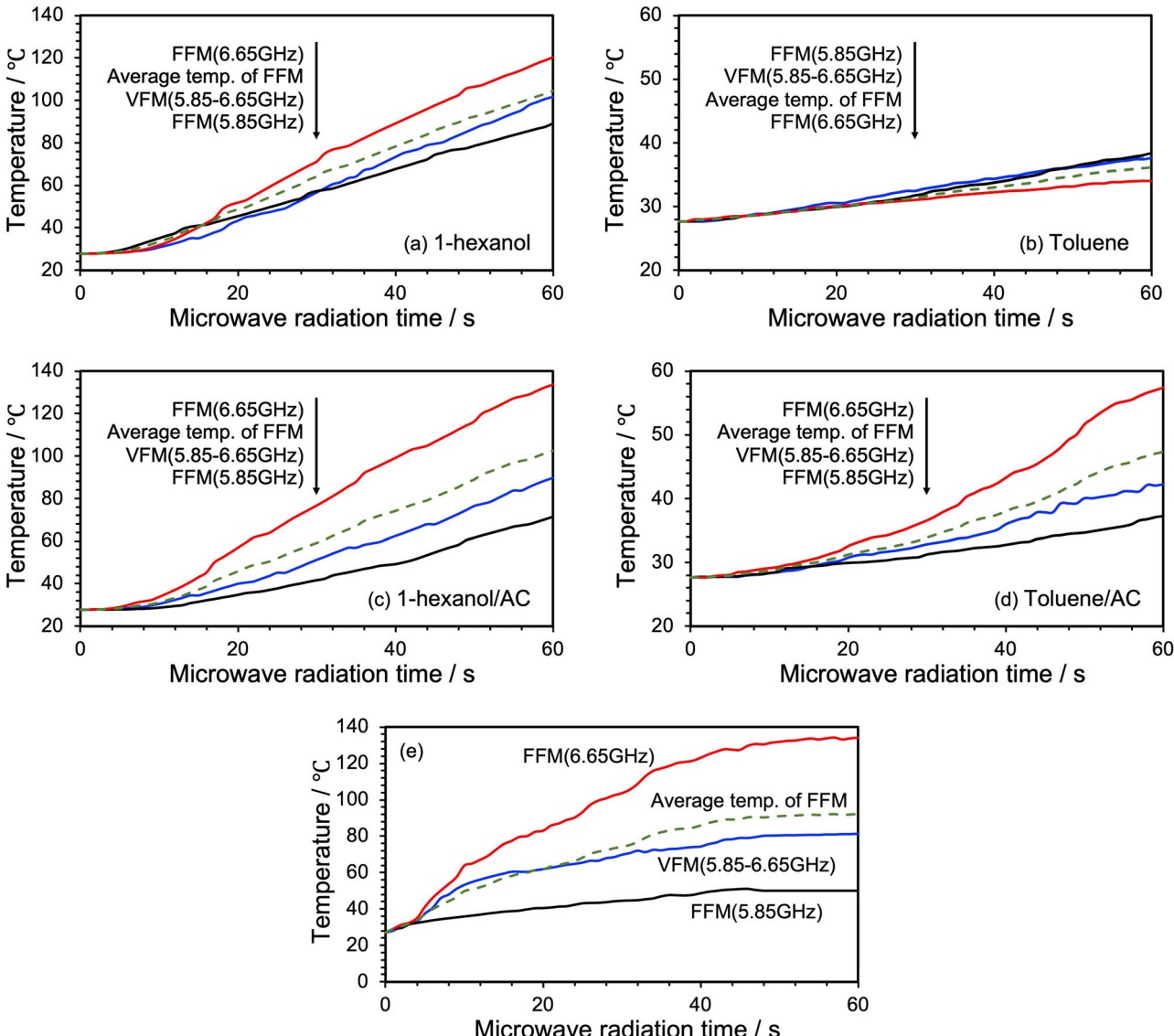

**Fig. 7 Temperature rise profile of 1-hexanol and toluene solvent with and without AC particles under VFM and FFM irradiation.** VFM irradiation (blue line), FFM irradiation (6.65 GHz: red line and 5.85 GHz: black line) and change in average temperature of two FFMs (green dotted line), **a** 1-hexanol, **b** toluene, **c** AC dispersed in 1-hexanol, **d** AC dispersed in toluene, **e** naked AC.

there appears to be a mismatch between the theoretical average temperatures of the two FFM irradiations and the temperatures of the VFM irradiation when sweeping the frequency at a constant rate. This mismatch is not due to the heat generated by the AC, but rather to some other source, most likely to the uniformity of the E-field intensity which is reduced under FFM irradiation as a result of the postulated discharge phenomena; that is to the generation of hot spots.

In order to substantiate the latter hypothesis, we did in fact observe electric discharge phenomena following microwave heating of an AC/toluene sample in a quartz test tube, especially under FFM irradiation; in particular, multiple discharges were observed near the AC surface under FFM irradiation with 6.65-GHz microwaves. To further demonstrate unambiguously these visually observed phenomena, we dispersed larger size AC particles (450–630 μm) in toluene (5 mL) in a quartz petri dish (inner diameter, 4.0 cm; height, 1.7 cm), and then proceeded to observe the dynamic change of the AC particles and generation of the electric discharges under both VFM and FFM irradiation (microwave power, 45 Watts) – results are displayed in Fig. 8.

VFM irradiation of AC particles in toluene solvent (Fig. 8a) revealed no change in the position of the AC particles. On the other hand, under FFM irradiation (Fig. 8b), AC particles began to move inside the solution immediately after irradiation, which was then followed by linear interconnections between the particulates. Furthermore, after 60 s, these linked particulates began to aggregate, and hot spots (discharge phenomena) began to be observed at multiple locations. After 120 s of FFM microwave irradiation, aggregation progressed further, and orange hot spots were observed at various locations. Note that the temperature of the orange light from such plasma can generally reach about 930 °C[33]. Clearly then, VFM irradiation not only warrants uniformity of the electric field inside the sample, but more importantly achieves suppression of hot spots.

A relevant question then is whether such discharge phenomena occur in polar solvents as well? Indeed they do, as illustrated in Fig. 8c, which reveals aggregation of AC particles and formation of hot spots under VFM irradiation of activated carbon dispersed in 1-hexanol. However, no hot spots were observed from the AC aggregates. Consequently, it seems that activated carbon had a

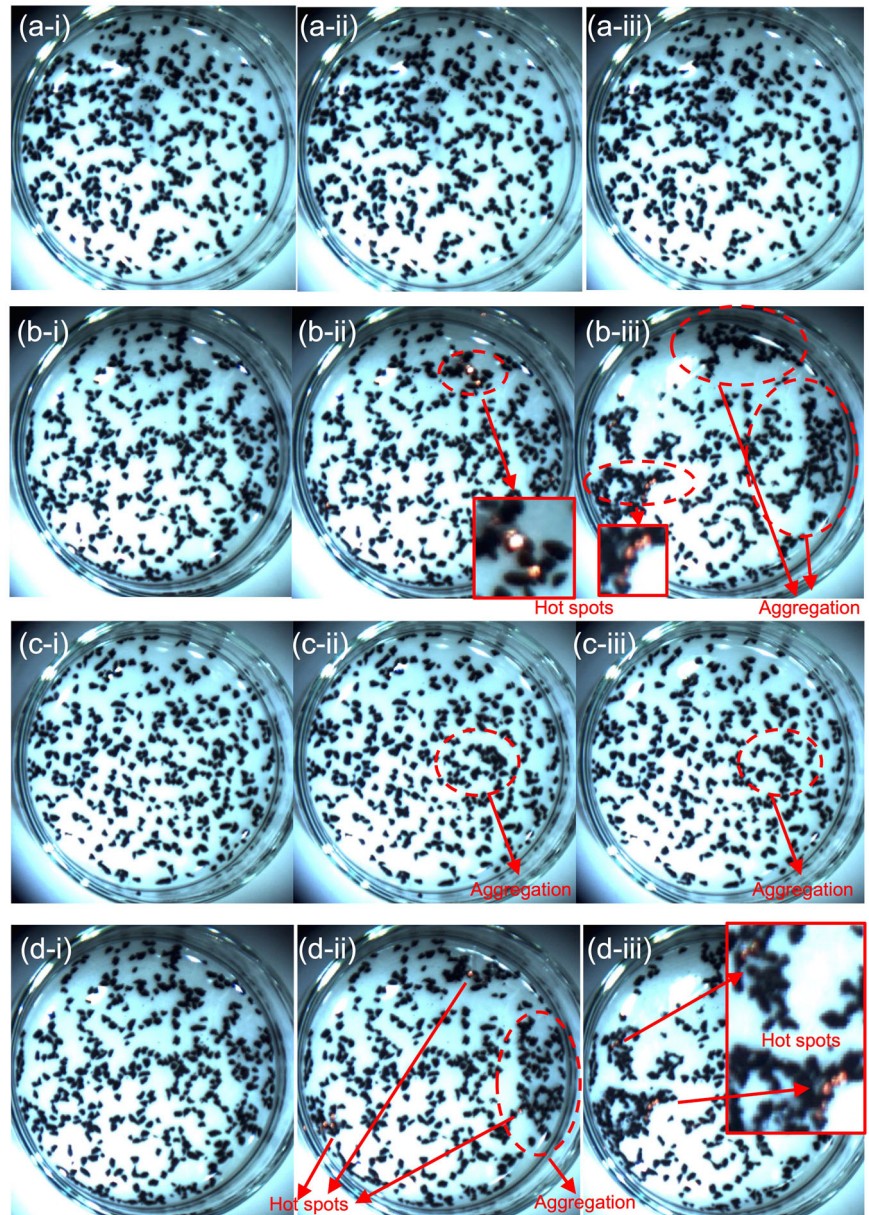

**Fig. 8 Photographs before and after AC heating in toluene and 1-hexanol solvents subjected to VFM irradiation (5.85 to 6.65 GHz) and FFM irradiation (6.65 GHz).** AC heating within **a**, **b** toluene and **c**, **d** 1-hexanol by **a**, **c** VFM irradiation (5.85 to 6.65 GHz) and **b**, **d** FFM irradiation (6.65 GHz). (i) initial sample, (ii) after a 60-s irradiation period, and (iii) after irradiation for 120 s.

low affinity for polar solvents and self-aggregated as a result of the physical attraction between AC particles. In contrast, under FFM irradiation (Fig. 8d), aggregation progressed rapidly, and orange hot spots were observed at multiple locations. Regardless of whether the solvent is polar or non-polar, VFM irradiation ensured uniform electric field distribution, thereby inhibiting the formation of hot spots. On the other hand, under FFM, the AC particles tended to aggregate and generate hot spots everywhere under the influence of the electric field, thus further contributing to heating of the solvent.

Although polarization inside the AC particulates is discontinuous (Fig. 9a) on application of an *E*-field, a perturbed *E*-field distribution and standing waves generated by FFM irradiation led charges to accumulate at the interface, which cause an apparent polarization that leads to microwave heating. As a result, the polarized activated carbon particles induce electrostatic coupling (Fig. 9b)[23]. On the other hand, the electrons ($e^-$) on the

protruding part of the AC surface are accelerated by the microwave *E*-field and migrate to the corner of another AC surface edge through a minute space between them (Fig. 9c); this creates an electrical discharge (hot spot). In other words, the AC surface edges behave like small antennas at which a high current may be established, thereby causing micro arcing between two antennas. Consequently, the probability of occurrence of discharges increases as the coupling of such AC particles develops. It appears then that in addition to self-heating of activated carbon caused by the microwaves under FFM irradiation, local high-temperature fields from the hot spots add to the heating of the solvent medium. As such, under VFM irradiation the temperature of the solvent is lower than the theoretical value.

**Yields in the scaled-up synthesis of 4-methylbiphenyl under VFM and FFM irradiation.** The model reaction we adopted to

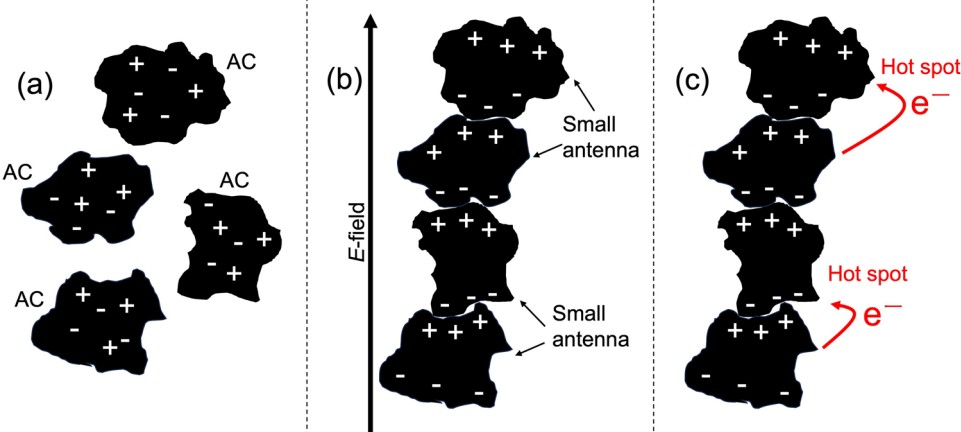

**Fig. 9 Cartoon images illustrating the polarization of AC particles (ACs) under microwave *E*-field conditions, and the corresponding electrostatic coupling. a** No microwave conditions and **b**, **c** under microwave conditions.

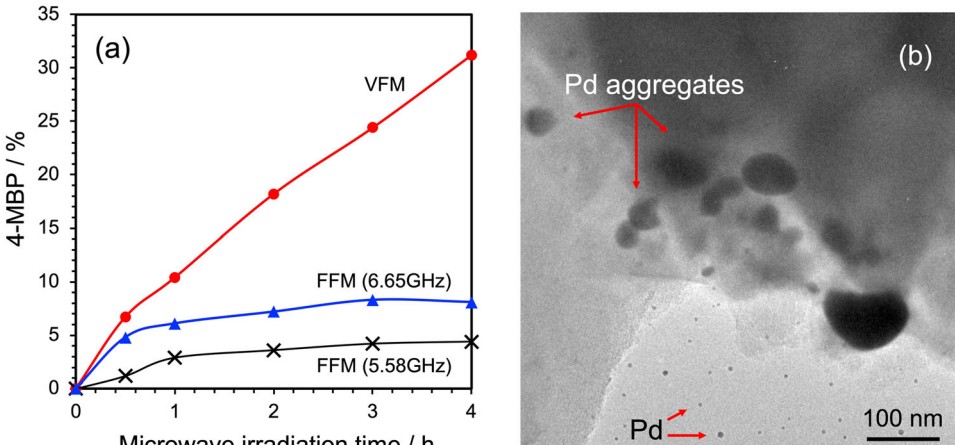

**Fig. 10 Product yields of 4-methylbiphenyl (4-MBP) in mixed toluene/1-hexanol solvent under VFM irradiation (continuously swept from 5.85 to 6.65 GHz in 0.1 s) and under irradiation with FFMs at 5.85 or 6.65 GHz; TEM image of the Pd/AC catalyst surface after 20 min of 6.65-GHz FFM microwave irradiation. a** Product yields of 4-methylbiphenyl (4-MBP) and **b** TEM image.

test our hypotheses involves the scaled-up synthesis of 4-methylbiphenyl (4-MBP) *via* the Suzuki-Miyaura cross-coupling reaction in a mixed toluene/1-hexanol solvent in the presence of Pd/activated carbon (Pd/AC) catalyst particles. The relevant synthetic yields of 4-MBP by VFM and FFM (5.85 GHz and 6.65 GHz) irradiation are reported in Fig. 10a. Microwave irradiation was controlled by proportional-integral-differential (PID) that maintained the reaction temperature of the solvent at ca. 110 °C.

Under VFM irradiation conditions, the synthesis yield of 4-methylbiphenyl increased in proportion to the synthesis time: yields of ca. 10.4% and ca. 31.2% were observed after 1 h and 4 h of irradiation, respectively. In contrast, the yield was 2.9% (1 h) and 4.4% (4 h) under 5.85-GHz FFM irradiation, and 6.1% (1 h) and 8.1% (4 h) under 6.65-GHz FFM irradiation conditions. FFM irradiation at 6.65 GHz and 5.85 GHz accelerated the synthesis initially, but then the Pd catalyst formed aggregates due to the heat generated by the discharges (hot spots) and caused loss of the catalytic activity of Pd/AC and arrested the formation of 4-MBP. Figure 10b displays a transmission electron microscopy (TEM; Hitachi Model U-331) image of AC particle surface after FFM (6.65 GHz) irradiation. The Pd nanoparticles on the AC surface dissolved under the influence of the discharges and subsequently formed Pd aggregates on the AC support, whereas no such events occurred or at least observed under VFM. Note

that the melting point of bulk-size palladium is 1554.9 °C, which decreases to ca. 223 °C for nano-sized icosahedral-shaped particulates[34]. Moreover, the synthesis of 4-MBP with VFM microwaves has been demonstrated even at a 5-fold scale-up in a 7.5-fold larger reactor compared to our earlier synthesis in a single-mode cavity[26]. Additionally, even under the latter scaled-up conditions, induced discharge phenomena on the surface of the Pd/AC catalyst was suppressed by VFM microwaves, establishing that the activity of the Pd catalyst was maintained.

## Concluding remarks

A serious and non-trivial problem in microwave-assisted organic syntheses is that microwaves are electromagnetic waves, which cause wavelength-dependent heterogeneity in the *E*-field distribution, as well as affect the resulting temperature uniformity upon sample heating. To resolve some of these issues, fundamental research on microwave cavities and reaction vessels has been conducted for some time. In some cases, microwave field irregularities cause the formation of hot spots (electrical discharges), especially in reactions involving solid catalysts. The present study has shown that such issues can be resolved by alternate microwave irradiation methods. Using an optical electric field sensor to measure and analyze the *E*-field distribution during actual microwave irradiation, we have shown that the

VFM irradiation method is critical in leveling the $E$-field distribution inside the cavity. In addition, to reconcile this fact, we have also demonstrated the characteristics of VFM irradiation in the heating of solvents and activated carbon particulates, as well as the effect(s) on the synthesis of 4-MBP. Accordingly, the results reported herein suggest that the VFM method is a useful technology for scaling up microwave chemistry at an industrial scale, thereby eliminating whatever obstacles microwave chemistry has hereto faced to enter the chemical industry.

## Experimental setup

**Microwave system**. High-speed sweeping of the microwave frequency used a VariWave system (Lambda Technologies Inc.) operated under irradiation with VFM microwaves generated from a semiconductor generator. The uniformity of the $E$-field with respect to VFM microwave irradiation was elucidated in the present study using rapidly swept frequencies over a frequency width from 5.85 to 6.65 GHz (wavelengths: 5.12 cm to 4.51 cm; sweep frequency range: 0.8 GHz); the frequency width was divided into 4096 discrete frequencies at 0.1 s intervals in such a way that the residence time for any of the 4096 frequency standing wave patterns was $25 \times 10^{-6}$ s (or 25 μs). The frequency switching width was ca. 195 KHz (= 0.8 GHz/4096 steps). Note that the 5.12-to-4.51 cm wavelengths of the VFM radiation are shorter than the 12.24-cm wavelength of the more common 2.45-GHz microwave heating frequency in the multimode cavity. The internal dimensions of the microwave multimode cavity (herein noted as the "cavity") were 35.6 cm (depth) by 29.2 cm (width) by 28.0 cm (height). The microwave irradiation port was installed on the side wall (rear wall) opposite the front lid of the cavity side at a position 19.2 cm from the bottom of the cavity (at the center of left and right positions). The internal dimensions of the irradiation window of the WRI-70 waveguide were 1.578 cm (width) by 3.485 cm (height). The block diagram of the VFM control system is displayed in Figure SI-1 (see Supplementary Information)[35].

For comparison, a conventional microwave chemical synthesis equipment was also used (Milestone General Co., Ltd.; Flexi-iWAVE system) that consisted of two 2.45-GHz magnetrons to generate the microwaves, together with a mode stirrer for uniform irradiation with the $E$-field in the multimode applicator, devised so as to change the intensity over time with the mode stirrer. The internal dimensions of the cavity were 43 cm (width) by 40 cm (depth) by 41 cm (height), with FFM irradiation being performed from the microwave irradiation port located in the upper portion of the apparatus.

**Synthesis of 4-methylbiphenyl (4-MBP)**. The synthesis of 4-methylbiphenyl was carried out using the Suzuki-Miyaura cross-coupling process so as to compare the current results with those from our earlier study[26]. The Pd/AC catalyst (1.40 g; mesh size of activated carbon (AC) support particulates = 0.65 mm), phenylboronic acid (0.96 mmol; 0.60 g), 1-bromo-4-methylbenzene (0.72 mmol; 0.60 g), $K_2CO_3$ as the base (1.40 mmol; 1.00 g), and mixed 1-hexanol/toluene as the solvent (60 mL; volume ratio, 1:1) were mixed under an Ar atmosphere and subsequently added to the quartz cylindrical reactor (internal diameter, 6.0 cm; height 15.0 cm). In contrast to our previous experiments that used a single-mode cavity[26], the synthesis of 4-MBP was carried out with a scale-up in mind using 5-fold greater sample quantities and a 7.5-fold larger reactor diameter, which was closed with a silicone plug connected to a silicon pipe protruding out of the microwave cavity. The quantity of Pd supported on the AC particulates was ca. 3.1 wt.% ascertained by atomic emission spectroscopy (Shimadzu ICPE-9000 apparatus).

Reaction yields of 4-methylbiphenyl were determined by gas chromatography (Shimadzu model 2014 apparatus equipped with a Shimadzu GLC Ultra alloy-1 capillary column; helium was the carrier gas; column temperature, 100–260 °C; heating rate, 20 °C min$^{-1}$) from samples suitably prepared from various dispersions. A pure sample of 4-methylbiphenyl (Fujifilm Wako Chemicals, Ltd., 100% GC standard) was used to calibrate the chromatograph.

**Optical electric field sensor**. The $E$-field intensity of the VFM and FFM irradiation microwaves was measured in the multimode cavity using an SH-10MS optical electric field sensor (sensor receiving area: 1 mm; fiber diameter: 2 mm) equipped with a crystal head (Seiko Giken Co., Ltd.) that was irradiated through the optical fiber cable using a semiconductor laser. The sensor head was fixed using Styrofoam at the position such that the $E$-field could be observed in the cavity (a photograph of the measurements apparatus is reported in Figure SI-2 of the Supplementary Information). Prior to this, however, we compared the $E$-field intensity at the same location in the cavity with and without Styrofoam. To the extent that there was no difference in electric field intensity between the two, we established that there was no effect from the Styrofoam. The sensor head is protected by polycarbonate ($15 \times 15$ x 70 mm$^3$). In addition, inasmuch as the amplitude of the laser light might change due to the microwave $E$-field, the $E$-field intensity was therefore measured by converting the amplitude of the laser light into a voltage. The measured value of the spectrum is given dBm units, which were then transformed into an $E$-field intensity (in units of dBμV m$^{-1}$) using an antenna correction factor (AF is the sensor eigenvalue given from the manufacturer; Seiko Giken Co., Ltd.) to convert dBm units into dBμV units. Note that the AF correction factors are unique to each device. The $E$-field intensity was ultimately converted into kV m$^{-1}$ units.

Since the sensor head has direct connectivity with respect to the incident direction of the $E$-field, we measured the $E$-field intensity in three axial directions: horizontal direction ($E_1$), depth direction ($E_2$), and vertical direction ($E_3$). These results were combined using Eq. 2 to estimate the $E$-field intensity (in units of kV m$^{-1}$) at position ($E$) within the cavity.

$$E[kV\ m^{-1}] = \sqrt{(E_1[kVm^{-1}])^2 + (E_2[kVm^{-1}])^2 + (E_3[kVm^{-1}])^2}$$
(2)

It is relevant to note that every time we finished measuring the electric field at one location, we shifted the field 2 cm and repeated the measurement. These operations were repeated twice and established that the difference was less than 2% after that. However, since a stable electric field could not be obtained for the FlexiWAVE system, the average value of no less than five measurements is reported in Fig. 3c-ii.

## Data availability

The data sharing in the Figures that support the findings of this study are available with the identifier https://doi.org/XXXXXXXX. The source data behind the graphs in the figures can be found in Supplementary Data 1.

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

## Acknowledgements
We wish to thank Mr. Osawa of Seikoh Giken. Co. Ltd. for assistance with the optical electric field sensors. We also wish to thank Lambda Technologies and Tokyo Instruments Inc. for assistance with the microwave VFM system. One of us (N.S.) is grateful to the staff of the PhotoGreen Laboratory at the University of Pavia (Italy) for their continued hospitality.

## Author contributions
S.H. designed and guided this project. H.M. conducted the experiments. S.H. wrote the first draft, which was subsequently re-examined following various discussions regarding the data's interpretation by N.S. and S.H.

## Competing interests
The authors declare no competing interest.

## Additional information

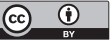

