## [Peer Review File · Communications Chemistry]

Reviewers' comments:

Reviewer #1 (Remarks to the Author):

The manuscript of prof. Horikoshi et al. is well written and can be accepted after minor revision.

Please consider the following comments:

Line 102:

A single-mode cavity has only one mode, if you changing the frequency you will generate also a other mode. If the mode will not change the position of the maximum and minimum of the electrical field will be at the same place -> you will have not a homogeneous heating.

You have two possibilities to get a homogeneous microwave heating:

Move the load.

Move the electrical feel (mechanical or electrical)

The goal is to get the same integrated microwave power at each point of the load.

Line 277:

The spectrum of a magnetron is changing with the level of the anode current (line pulling) and the load / stirrer (load pulling). If you using a DC anode voltage the spectrum of a Magnetron will be also smaller. With a DC anode voltage, you can increase the lifetime of a Magnetron but it will possible that you reduce the homogenous of the electrical feel and the heating.

Line 306:

I'm not sure if I understood the point correct.

Inside a cavity and a waveguide, you have always a cavity resonator, with the resonance frequency you get a electrical mode. Yes it will possible to have different modes at the same time, but this modes will overlies to one new mode. With sweeping the frequency you will generate a new overlies mode. And as you see at Figure 3, with a bigger frequency sweep range you will have a more different overlies modes.

Line 460:

It will be also possible that you generate some small antennas where you get a high current on it or micro arcing between two antennas.

Line 513:

The VariWave-System, do you have a datasheet or manual of this unit, because I know them but never get real information how the sweeping the frequency. The only what I have is they work with a standard magnetron 2.45GHz and overlies a low power signal with different frequencies which will be not so effective as a amplifier.

General comments:

At the last 25 years I had some projects with amplifiers, which can sweep between 2.4 and 2.5GHz (GSM-band), I can confirm with a frequency sweep you will get a homogeneous microwave heating and can save up to 30% microwave power. It will be also possible to heat up inside a mixture of different material one material more as the other.

Reviewer #2 (Remarks to the Author):

In this paper, the authors reported the experimental in-situ three-dimensional measurements of the E-field's uniform distribution of Variable Frequency Microwaves (VFM) within a multimode cavity under high power conditions, and subsequent comparison to fixed frequency microwaves (FFM).

The paper is very interesting, properly divided in sections and sub-sections. It needs minor corrections before being considered for publication in the journal.

- At line 207 the authors wrongly referred to figure 2: perhaps they meant figure 3?
- In the authors idea, which could be a potential industrial application of the proposed VFM? Have the authors an idea of the costs of this device if compared to a classical microwave generator? May the authors add a comment?
- Some errors are present in the legend of some figures, please check.

Reviewer #3 (Remarks to the Author):

The present study focus on the problem of heterogeneous heating of microwave irradiation and investigated the performance of a Variable Frequency Microwaves method by employing thermopaint to show the temperature distributions as well as a novel commercial probe that can demonstrate three-dimensional electrical intensity. The experimental results showed the advantages of Variable Frequency Microwaves, including the depression of electrical discharges. Considering this study is helpful to the scale-up of microwave technology, I would like to recommend this paper should be accepted by Commun. Chem. after minor revision. The comments are exhibited as follows.

(1) Page 4: More details of the used optical electric field sensor should be provided in Figure 1, such as the shape, size and location of the probe.

(2) Page 6: According to the measurement results reported in many literatures and our previous work, the dielectric loss of hexanol is nearly 0.8-0.9, much lower than the values in Figure 6. The authors are recommended to supply more details of the network that they used and check the values, carefully.

(3) The authors should polish their English writing as a large amount of sentences are filled with colloquialism. Besides, some details should be checked. For example, it should be 20 cm × 20 cm or 20 × 20 cm², rather than 20 x 20 cm at Page 4 (Line 140). This mistake also appears in Figure 1.

Reviewers' Comments and Responses

Corrections are indicated with yellow markers in the manuscript.

Reviewer #1 (Remarks to the Authors):

The manuscript of prof. Horikoshi et al. is well written and can be accepted after minor revision. Please consider the following comments:

Line 102:

A single-mode cavity has only one mode, if you changing the frequency you will generate also a other mode. If the mode will not chance the position of the maximum and minimum of the electrical field will be at the same place -> you will have not a homogeneous heating.

You have two possibilities to get a homogeneous microwave heating:

- Move the load.
- Move the electrical feel (mechanical or electrical)

The goal is to get the same integrated microwave power at each point of the load.

We thank the reviewer for this information. Accordingly, some changes were made to the text in section 2.2.1

Line 277:

The spectrum of a magnetron is changing with the level of the anode current (line pulling) and the load/stirrer (load pulling). If you using a DC anode voltage the spectrum of a Magnetron will be also smaller. With a DC anode voltage, you can increase the lifetime of a Magnetron but it will possible that you reduce the homogenous of the electrical feel and the heating.

We thank the reviewer for enlightening us on this engineering aspect.

Line 306:

I'm not sure if I understood the point correct. Inside a cavity and a waveguide, you have always a cavity resonator, with the resonance frequency you get a electrical mode. Yes it will possible to have different modes at the same time, but this modes will overlies to one new mode. With sweeping the frequency you will generate a new overlies mode. And as you see at Figure 3, with a bigger frequency seep range you will have a more different overlies modes.

This question refers to the VariWave equipment we used. Please note that this device was developed by Lambda Technologies, and it was not our intention in this paper to discuss such

details.

Line 460:

It will be also possible that you generate some small antennas where you get a high current on it or micro arcing between two antennas.

We thank the reviewer. The paragraph involving Figure 9 together with Figure 9 have been updated.

Line 513:

The VariWave-System, do you have a datasheet or manual of this unit, because I know them but never get real information how the sweeping the frequency. The only what I have is they work with a standard magnetron 2.45GHz and overlies a low power signal with different frequencies which will be not so effective as a amplifier.

We obtained the system diagram for generator and posted its contents in section 4.1. However, the purpose of this paper was not to develop a device. Please note that we utilized a commercially available equipment that is also available to everyone else.

General comments:

At the last 25 years I had some projects with amplifiers, which can sweep between 2.4 and 2.5 GHz (GSM-band), I can confirm with a frequency sweep you will get a homogeneous microwave heating and can save up to 30% microwave power. It will be also possible to heat up inside a mixture of different material one material more as the other.

We thank the reviewer for this very useful comment

Reviewer #2 (Remarks to the Authors):

In this paper, the authors reported the experimental in-situ three-dimensional measurements of the E-field's uniform distribution of Variable Frequency Microwaves (VFM) within a multimode cavity under high power conditions, and subsequent comparison to fixed frequency microwaves (FFM). The paper is very interesting, properly divided in sections and sub-sections. It needs minor corrections before being considered for publication in the journal.

- At line 207 the authors wrongly referred to figure 2: perhaps they meant figure 3?

Thank you very much for this suggestion. Indeed it was a confusing expression, and thus the

equipment condition has been changed to mean the same as in Figure 2b-ii.

- In the authors' idea, which could be a potential industrial application of the proposed VFM? Have the authors an idea of the costs of this device if compared to a classical microwave generator? May the authors add a comment?

We added our thinking on this point to section 2.2.2 (III).

- Some errors are present in the legend of some figures, please check.

Figures 4 and 6 have now been updated.

Reviewer #3 (Remarks to the Authors):

The present study focus on the problem of heterogeneous heating of microwave irradiation and investigated the performance of a Variable Frequency Microwaves method by employing thermopaint to show the temperature distributions as well as a novel commercial probe that can demonstrate three-dimensional electrical intensity. The experimental results showed the advantages of Variable Frequency Microwaves, including the depression of electrical discharges. Considering this study is helpful to the scale-up of microwave technology, I would like to recommend this paper should be accepted by Commun. Chem. after minor revision. The comments are exhibited as follows.

(1) Page 4: More details of the used optical electric field sensor should be provided in Figure 1, such as the shape, size and location of the probe.

Details and Figure 11 have been added in section 4.3.

(2) Page 6: According to the measurement results reported in many literatures and our previous work, the dielectric loss of hexanol is nearly 0.8-0.9, much lower than the values in Figure 6. The authors are recommended to supply more details of the network that they used and check the values, carefully.

We greatly appreciate this reviewer's comments. The values pointed out by the referee are reasonable compared to the values we have already reported. We re-measured using another network analyzer with precisely calibration of equipment. The data showing the re-measured values are now shown as Figure 6.

(3) The authors should polish their English writing as a large amount of sentences are filled with

colloquialism. Besides, some details should be checked. For example, it should be $20\text{ cm} \times 20\text{ cm}$ or $20 \times 20\text{ cm}^2$, rather than $20 \times 20\text{ cm}$ at Page 4 (Line 140). This mistake also appears in Figure 1.

Units have been modified to read cm^2 . Moreover, we have reread the whole paper and where necessary we made some changes with the view of avoiding the use of colloquialisms.

REVIEWERS' COMMENTS:

Reviewer #1 (Remarks to the Author):

The Authors answered to all the amendments, now the revised version of the manuscript can be accepted for publication

Reviewer #2 (Remarks to the Author):

The authors well replied to the comments of the reviewer, but one minor issue still remains: At the beginning of section 2.2 the authors referred to figure 2b-ii. Which is this figure?

Reviewer #3 (Remarks to the Author):

In view of the corrections already made, that is available on the site. The revised manuscript has significantly improvement than original. The revision has sufficiently replied to the question and makes detailed supplement and discussion. Therefore, the article is recommended for publication in the Communications Chemistry.

Reviewers' Comments and Responses

Reviewer #1 (Remarks to the Authors):

Reviewer #1 (Remarks to the Author):

The Authors answered to all the amendments, now the revised version of the manuscript can be accepted for publication.

We thank the reviewer for the positive comment.

Reviewer #2 (Remarks to the Author):

The authors well replied to the comments of the reviewer, but one minor issue still remains: At the beginning of section 2.2 the authors referred to figure 2b-ii. Which is this figure?

We thank this reviewer for pointing out the oversight. Figure 2b-ii has now been corrected to Figure 1b-ii.

Reviewer #3 (Remarks to the Author):

In view of the corrections already made, that is available on the site. The revised manuscript has significantly improvement than original. The revision has sufficiently replied to the question and makes detailed supplement and discussion. Therefore, the article is recommended for publication in the Communications Chemistry.

We thank the reviewer for the positive comment.